# Improvement of German Chamomile (*Matricaria recutita* L.) for Mechanical Harvesting, High Flower Yield and Essential Oil Content Using Physical and Chemical Mutagenesis

**DOI:** 10.3390/plants11212940

**Published:** 2022-11-01

**Authors:** Yasser E. Ghareeb, Said S. Soliman, Tarek A. Ismail, Mohammed A. Hassan, Mohammed A. Abdelkader, Arafat Abdel Hamed Abdel Latef, Jameel M. Al-Khayri, Salha M. ALshamrani, Fatmah A. Safhi, Mohamed F. Awad, Diaa Abd El-Moneim, Abdallah A. Hassanin

**Affiliations:** 1Genetics Department, Faculty of Agriculture, Zagazig University, Zagazig 44511, Egypt; 2Horticulture Department, Faculty of Agriculture, Zagazig University, Zagazig 44511, Egypt; 3Department of Botany and Microbiology, Faculty of Science, South Valley University, Qena 83523, Egypt; 4Department of Agricultural Biotechnology, College of Agriculture and Food Sciences, King Faisal University, Al-Ahsa 31982, Saudi Arabia; 5Department of Biology, College of Science, University of Jeddah, Jeddah 21959, Saudi Arabia; 6Department of Biology, College of Science, Princess Nourah bint Abdulrahman University, Riyadh 11671, Saudi Arabia; 7Department of Biology, College of Science, Taif University, Taif 21944, Saudi Arabia; 8Department of Plant Production, (Genetic Branch), Faculty of Environmental and Agricultural Sciences, Arish University, El-Arish 45511, Egypt

**Keywords:** gamma rays, German chamomile, *Matricaria recutita*, mechanical harvesting, mutations, oil content, oil quality, sodium azide

## Abstract

Chamomile (*Matricariarecutita* L.) is one of the most important medicinal plants with various applications. The flowers and flower heads are the main organs inthe production of essential oil. The essential improvement goals of chamomile are considered to be high flower yield and oil content, as well asthe suitability for mechanical harvesting. The present study aimed to improve the flower yield, oil content and mechanical harvestability of German chamomile via chemical and physical mutagens. Three German chamomile populations (Fayum, Benysuif and Menia) were irradiated with 100, 200, 300 and 400 Gray doses of gamma rays, as well as chemically mutagenized using 0.001, 0.002 and 0.003 mol/mL of sodium azide for 4 h. The two mutagens produced a wide range of changes in the flowers’ shape and size. At M_3_ generation, 18 mutants (11 from gamma irradiation and 7 from sodium azide mutagenization) were selected and morphologically characterized. Five out of eighteen mutants were selected for morphological and chemical characterization for oil content, oil composition and oil quality in M_4_ generation. Two promising mutants, F/LF5-2-1 and B/HNOF 8-4-2, were selected based on their performance in most studied traits during three generations, as well as the high percentage of cut efficiency and a homogenous flower horizon, which qualify them as suitable candidates for mechanical harvesting. The two mutants are late flowering elite mutants; the F/LF5-2-1 mutant possessed the highest oil content (1.77%) and number of flowers/plant (1595), while the second promising B/HNOF 8-4-2 mutant hada high oil content (1.29%) and chamazulene percentage (13.98%) compared to control plants. These results suggest that the B/HNOF 8-4-2 and F/LF5-2-1 mutants could be integrated as potential parents into breeding programs for a high number of flowers, high oil content, oil composition and oil color traits for German chamomile improvement.

## 1. Introduction

*Matricaria* L., a member of the Asteraceae family, is considered one of the most widely used medicinal plants around the world. Its origins can be traced back to the near east as well as south and east Europe. It is a diploid (2n = 18) plant that is frequently outcrossing [1,2]. For thousands of years, chamomile has been employed in herbal treatments, and it was known in ancient Egypt, Greece and Rome [3]. It is one of Germany’s most important crops for pharmaceutical and cosmetic reasons, in addition to being a raw material for the food industry [4]. The essential oil is extracted from the inflorescences by steam distillation or solvent extraction, with yields ranging from 0.24 to 1.90% [5]. Coumarin, flavonoids, -bisabolol, -bisabolol oxide A and B, chamazulene, sesquiterpenes, spiroethers and other components such as tannins, anthemic acid, choline, polysaccharides and phytoestrogens are the active principles in German chamomile (*Matricaria recutita* L.) [6]. The flower of chamomilecontains essential oils from 0.2 to 1.9% [7], which have various uses. The primary uses of chamomile are as an anti-inflammatory and antiseptic, and it is also antispasmodic and mildly sudorific. Internally, chamomile is used as a tisane for pain-related stomach issues, slow digestion, diarrhea and nausea. Less frequently, but just as well, it is used for urinary tract irritation and painful menstruation [8]. The drug of chamomile in powder form is used externally forwounds slow to heal, skin eruptions and infections, such as shingles and boils, as well as for hemorrhoids and inflammation of the mouth, throat and the eyes.

Unfortunately, chamomile genotypes are extremely limited, and the majority of chamomile varieties are either landraces or populations [9,10]. As a result, the drive to discover new sources of diversity is crucial. Increased genetic diversity and new character induction were used to actively conduct mutation breeding for crop breeding [11]. The physical and chemical mutagens were artificially used to induce mutations, which increases the mutation frequency when compared to the spontaneous occurrence. However, excellent production efficiency is required for the widespread use of these mutants in plant breeding [12]. With the use of induced mutagenesis, it is possible to improve economic and quality traits for crop improvement in a short time [13]. Chamomile mutation breeding succeeded in the development of promising chemo type CIM–Ujjwala in India using a 10–100 kilo rad gamma ray. This mutant possessed a high dry flower yield and high essential oil content with a light brown oil color in M_4_ plants [14].

The harvesting process is a sharp problem in chamomile production since it results in significant losses in flower yield and quality. On hot days, chamomile is harvested when the majority of the inflorescences and the ligulate flowers are opened and disposed of horizontally (after 9 Am), while the tubular ones are flowering with a yellow color and a greenish-yellow coloring. When the ratio of blossoms to opened flower heads is 1:1, the volatile oil content is at its greatest quantity [15]. As a result, the harvesting time and mode have a significant impact on the quality of the plant material. Chamomile inflorescences are harvested by pulling them either manually or mechanically; manual harvesting is performed by qualified personnel and results in high-quality products; otherwise, product quality is uneven, however, it can be costly and inefficient in Europe [16]. Chamomile blooms multiple times a year, however, it is only harvested once, twice or three times [17]. In Egypt, chamomile is considered a main medicinal plant; its production is divided into old land (delta) of 14,022 acres and land outside 2545 acres [18], but most areas are divided into small farms (0.5–2 Acres) and, subsequently, are harvested manually four times due to it being not uniform, which is highly expensive. In addition, chamomile cultivation in the desert at a large scale was harvested by mechanical harvesters two times only, and subsequently, lost a lot of flower yield because chamomile flowers blossomat different times. Surprisingly, an Egyptian cultivar or variety with ahigh content of bisabolol and chamazulene has not been released yet and a claim “for the future it is important to use selected seeds with a high percentage of azule” remains in force for a very long time [5]. Therefore, the present study aimed to induce mutations of three German chamomile populations from the three chamomile cultivation regions to produce mutants suitable for mechanical harvesting in two or three stages of a flower blossom, as well as high productivity and oil content.

## 2. Results

### 2.1. Characterization of M_2_Populations

The air-dried seeds of the three populations were irradiated with 100 Gy, 200 Gy, 300 Gy and 400 Gy of gamma rays. The dose rate was 5.6 Gy/minute. In a separate experiment the seeds were treated with sodium azide (NaN_3_) with 0.001, 0.002 and 0.003 mol/mL of SA for four hours. In the M_2_ population, 1180 plants were screened for the mutagenicity effects of gamma rays and 810 1180 plants were screened for the mutagenicity effects of sodium azide. M_2_ populations that contained various morphological mutants were obtained from the three populations of German chamomile. These mutants involved traits affecting plant height traits (tall, dwarf, semi-dwarf), large stem diameter, the high number of branches, early flowering, late flowering and the high number of flowers (Table 1). Treatments with gamma rays and sodium azide mutagens produced a varied number of mutations; 37 mutants were produced for plant height (dwarf, semi-dwarf and tall), 38 mutants involved traits affecting early and late flowering, in addition to 25 mutants with a high number of flowers. Results showed that low doses of gamma rays (100 Gy and 200 Gy) produced higher mutation frequency than higher doses (300 Gy and 400 Gy). In contrast, higher concentrations of sodium azide (0.003 mol/mL and 0.002 mol/mL) gave higher mutation frequency while the lower concentration (0.001 mol/mL) produced lower mutation frequency. We also found that gamma rays had a lower mutation frequency (15.94%) than the mutation frequency scored by sodium azide (16.2%). 

The selected M_2_ mutants, i.e., fresh weight of flowers per plant, dry weight of flowers per plant, number of flowers per plant, plant height, stem diameter, number of branches per plant and days to flowering were morphologically characterized (Appendix A). The dwarf and semi-dwarf mutants have no good yield characters except the fourth mutant of Fayoum (F/SD4) and Menia (M/SD4) populations. It is also notable that the high number of flower mutants and large stem diameter mutants possessed high yield traits in the three populations. The results also showed that most tall plants and late flowering mutants possessed high yield characteristics. 

To reduce the blossoming stages for increasing mechanical harvesting efficiency, the selection of mutations suitable for mechanical harvesting using cut efficiency percentage was applied on M_2_ populations (Appendix A). The results recorded on M_2_ plants (Appendix A) indicated that mutations obtained from Benysuif populations were more suitable for mechanical harvesting than the Menia and Fayoum populations because of late flowering compared to Menia and Fayoum populations. The best kind of mutations suitable for mechanical harvesting were late flowering mutants followed by the high number of flowers followed by tall plants and the high number of buds, due to the sum of both blossoming stages (three and four) making up more than 95% of total flower yield. Dwarf, semi-dwarf and early flowering mutants were not suitable for mechanical harvesting. 

### 2.2. Characterization of M_3_ Mutants

The performance stability of selected M_3_ mutants from different classes of the populations were evaluated for numerous morphological features, oil content and oil colors (Table 2). The results showed that the late flowering mutants possessed the highest values for flower dry weight and number of flowers in all three populations. The results also showed that the mutants of the high number of flowers possessed high values for flowers’ dry weight and the number of flowers in the two populations Benysuif and Menia (Table 2).

In the Fayoum population, the F/LF 5-2 mutant showed the highest performance in morphological features and oil content (1.77%) and the oil colors were very light blue, so it is considered a desirable mutant (Table 2). In the same context, in the Benysuif population, the B/HNOF8-4 mutant possessed the highest performance in all studied traits except the number of branches and days to flowering, and the oil color was very light blue. Meanwhile, in the Menia population, M/HNOF 4-1 presented the highest performance in all studied traits except stem diameter, the number of branches and days to flowering. The results of M_3_ generation also confirmed that early flowering mutants showed stability in contrast to days to flowering, which showed low stability in Benysuif and Menia populations.

Economically, the traits of the number of flowers and oil content are of great importance to be selected during mutation breeding of German chamomile populations. In the Fayoum population, the F/LF 5-2 mutant possessed the highest values of those characteristics. In the Benysuif population, B/HNOF 4-3 and B/HNOF 8-4 mutants were the best, while in the Menia population, the early flowering mutants M/EF 4-1 and M/EF 5-2 performed the same. 

### 2.3. Characterization of M_4_ Mutants

Five promising M_4_ mutants, F/LF 5-2-1, F/HNOF3-1-1, B/HNOF 4-3-1, B/HNOF8-4-2 and M/HNOF 4-1-1 (Figure 1), were selected from M_3_ populations to evaluate their stability and their homogeneity regarding morphological characteristics, oil content, oil colors (Table 3) and cut efficiency percentage (Figure 2). The results showed that the five promising mutants in M_4_ generation presented close values to M_3_ generation, especially B/HNOF 8-4-2, M/HNOF4-1-1, F/LF5-2-1 and B/HNOF4-3-1, respectively. 

The results also showed that the five promising mutants gave a similar cut efficiency percentage at M_3_ and M_4_ generations and were suitable for mechanical harvesting in three stages, in other words, reducing blossoming stages from four to three stages (Figure 2). B/HNOF 8-4-2 and F/LF5-2-1, out of the five promising mutants evaluated in M4, possessed the highest values of flower fresh weight (g), flower dry weight (g), number of flowers, plant height (cm), stem diameter (mm), number of branches, days to flowering, oil content (%) and oil colors.

### 2.4. Essential Oil Composition Analysis of M_4_ Selected Mutants 

Gamma irradiation and sodium azide not only affected the oil content but also the oil composition of chamomile essential oil in the five M_4_ promising mutants. The basic composition of chamomile essential oil was recorded in Table 4. A total of 46 compounds were identified, which accounted for 95.46–100% of the total amount of oil. The main constituents found in the essential oils as detected by GC-MS were bisabolol oxide A (33.19–47.32%), bisabolone oxide A (1.34–12.36%), bisabolol oxide B (1.2–20.62%) and chamazulene (1.58–13.98%). The oil composition of the studied mutants was quite different, whereas other components appeared in amounts less than 2%. Generally, the examined mutants were clustered in two main groups: one concerning chamazulene and the second one concerning high α-bisabolol oxide A, B and chamazulene content. The analysis of the essential oil constituents revealed a low content of bisabolol and chamazulene in the M/HNOF 4-1-1 mutant where the chamazulene was absent, while in the F/LF 5-2-1mutant, the yield of bisabolol and chamazulene was equal or even surpassed the respective of the control. Concerning the percentage of bisabolol, mutant B/HNOF 4-3-1 outyielded all other mutants, estimated at 53.81%, followed by F/HNOF 3-1-1 (52.58). For chamazulene, the B/HNOF 8-4-2 mutant exhibited the greatest value (13.93%), followed by the F/HNOF 3-1-1 mutant (2.74%). The control presented 47.05% and 1.58% for bisabolol and chamazulene, respectively. It is evident that the highest chamazulene percentage, 13.98%, was achieved by the B/HNOF 8-4-2 mutant.

The late flowering elite mutant (F/LF5-2-1) possesses the highest oil content (1.77%) compared to control plants (0.9%) and the number of flowers/plant (1595) versus control plants (1075). The other promising mutant (B/HNOF 8-4-2) has high oil content (1.29%) against control plants (0.9%) as well as high chamazulene percentage (13.98%) compared to control plants (1.58%). These two promising mutants considered more suitable for mechanical harvesting are shown in Figure 1. 

## 3. Discussion

It has been established that both radiation and chemical mutagens (gamma rays and sodium azide) play a role in enhancing genetic variations of studied characters in German chamomile. Total mutations, also known as heritable changes to the genetic material, can occur at the chromosomal level or as point mutations [19]. It has been found that several chemical and physical mutagens cause variability for economic features in various crops [14,20]. In programs to improve various crop plants through mutation breeding, useful morphological mutations are crucial. Numerous researchers have concluded that gene mutations and chromosomal aberrations are the causes of morphological mutations [21].The morphological mutations concerning flowers’ fresh weight, flowers’ dry weight, number of flowers/plant, plant height, stem diameter, number of branches, days to flowering, oil content % and oil colors were noticed in the M_2_ generation of the three populations of German chamomile. Even though the majority of morphological mutants are not economically viable, many of them can be useful and utilized in crossbreeding programs or to improve quantitative traits, track crop evolution and conduct gene mapping investigations [22,23,24,25]. The frequency of various mutant types may be caused by various mutagens and treatment procedures [26]. The pleiotropic effects of the defective genes lead to morphological mutations [27]. The three populations of German chamomile varied in the frequency of morphological mutations. In contrast to Benysuef and Menia populations, the Fayoum population displayed a higher frequency of morphological mutants, demonstrating the inter-ecotype response to various doses and concentrations of physical and chemical mutagens. A report on the inter-varietal response to mutagen treatments was made by Gottschalk [27] in barley. The results of the current investigation showed that the M_2_ populations of chamomile contained tall mutants. Additionally, tall mutants were noted by Solanki, et al. [28] in *Lens culinaris* Medik. and Kumar, et al. [29] in Mungo L. using different mutagens. Dwarf mutants observed in the present study had short internodes, which could be due to a reduction in cell number and cell length. These dwarf mutants have also been reported in *Vigna mungo* L. Hepper [30], grasspea (*Lathyrus sativus* L.) [31] and barley (*Hordeum vulgare* L.) [32]. The dwarf mutants resulted due to a decrease in internode number or internode length. A decline in plant height may be caused by altered gibberellic acid or uneven mitotic divisions, according to several studies [33,34]. In wheat, the semi-dwarf mutant character is controlled by polygenes [35]. According to numerous researchers, many morphological mutations, such as tallness and dwarfism, are monogenic and recessive [36,37,38]. 

Based on our results, it is clear that gamma rays and sodium azide mutagens induced high variations in morphological features, essential oil content, fresh and dry flower yield and flowering periods depending upon the two mutagen types. Additionally, it was noted that the two mutagens also caused changes in oil composition and oil color in several mutants, including dark blue, light blue, dark yellow and light brown.

The cut efficiency percentage at M_3_and M_4_generations for all mutants also confirms the previous results that the late flowering mutants promise great results in the third and fourth cuts. These results are confirmed by the work of Albrecht et al. [39] who studied the breeding of a new variety of chamomile to increase the blossom product yield in up to three harvest stages through a homogenous flower horizon (pick height) and even flowering time. 

We conclude from the previous results that most late flowering and the high number of flower mutants at the M_4_generation possess high performance in addition to high oil content. These results are confirmed by Lal, et al. [14] who produced a new variety of chamomile to increase flower yield and essential oil content.

The results showed that the five promising mutants were found to be similar in cut efficiency percentage at M_3_ and M_4_ generations. Thus, they are suitable for mechanical harvesting in three stages. 

Interestingly, as a result of greater mutagenic efficacy and massive screening in field trials especially, the high percentage of cut efficiency and a homogenous flower horizon for the selected mutants led to reducing the significant loss in flower yield, which makes them suitable for mechanical harvesting, especially the late flowering elite mutant (F/LF5-2-1) that possesses the highest oil content and number of flowers, and the B/HNOF 8-4-2 mutant that possesses high oil content as well as high chamazulene percentage.

Despite the possibility that this study will support traditional breeding programs [40] for the genetic improvement of German chamomile, more recent genetic approaches such as genetic engineering [41], genome editing approaches [42,43], molecular markers [44,45,46,47] and phylogeny analysis [48,49,50,51] are important to determine the genetic diversity among different species, which could effectively be utilized to breed and develop chamomile for many desirable traits.

## 4. Materials and Methods

### 4.1. Plant Material

Seeds of three chamomile populations *(Matricaria recutita* L.) were collected from farmers of aromatic and medicinal plants, from Fayoum, Benysuif and Menia governorates, Egypt, of a local accession of German chamomile. This investigation was conducted during four successive winter seasons, 2018, 2019, 2020 and 2021, at the experimental farm of the Department of Genetics Faculty of Agriculture, Zagazig University, Egypt (Latitude: 30°35′15″ N, Longitude: 31°30′07″ E, Elevation above sea level: 16 m = 52 ft).

### 4.2. Mutagen Agents 

The seeds of the different populations of chamomile were treated with gamma radiation and sodium azide (NaN_3_) as follows:

#### 4.2.1. Physical Mutagen 

The air-dried seeds of each population were irradiated with 100 Gy, 200 Gy, 300 Gy and 400 Gy of gamma rays with a radioisotope Co^60^ source (Gamma chamber Model-900 supplied by Nuclear Research Center, Inshas, Egypt). The dose rate was 5.6 Gy/minute. 

#### 4.2.2. Chemical Mutagen

Separately, the seeds were treated with sodium azide (NaN_3_); one gram of seeds of each population was treated with 0.001, 0.002 and 0.003 mol/mL of SA for four hours with intermittent shaking at room temperature, washed for an hour under running water, then immediately sown. 

### 4.3. Agronomic Practices and Data Collection

As presented in Figure 3, the irradiated, as well as sodium azide, treated seeds of each population with control seeds were sown individually in rows in an incubator in the middle of August. Forty-five-day-old seedlings were transplanted in the open field.Surface irrigation was supplied every 3 weeks and after each cut. Routine agricultural practices were carried out as usually practiced in chamomile cultivation. During the seedling stage and before the flowering stages, weeds were pull out by hand. German chamomile is not a demanding crop in terms of fertilization. The plants were harvested separately in each treatment. M_3_ seeds from M_2_ selected mutants were grown in three rows with three replications for M_3_ generation, as well as the control of each population. In the 2021/2022 season, the seeds of M_4_ plants from selected mutants were sown in three replications with their parents to evaluate mutants. 

### 4.4. Measured Parameters

For the M_3_ and M_4_ generations, three replications of 30 seedlings each were sown for every treatment (90 plants/treatment) in rows and controlled in every population in a randomized complete block design (RCBD). The spacing was maintained at 50 between seedlings and 75 cm between rows. Data were collected from M_3_ and M_4_ for morphological and phenological traits including fresh weight of flowers per plant(g), dry weight of flowers per plant(g), number of flowers/plant, plant height (cm), stem diameter (mm), number of branches, days to flowering, oil content (%) and oil colors in three replicates. The number of flowers/plant was collected from four harvesting cuts (four cuts every 20 days during January, February and March). The flower cut is performed when the petals of the flowers are in horizontal position and dried under shade conditions for 7–10 days. The percentage of flower cut efficiency of all stages of harvest for the M_4_ mutants was calculated by the following formula: $$\mathrm{Flower}\;\mathrm{cut}\;\mathrm{efficiency}=\frac{\mathrm{Number}\;\mathrm{of}\;\mathrm{flower}\;\mathrm{per}\;\mathrm{each}\;\mathrm{cut}}{\mathrm{Total}\;\mathrm{number}\;\mathrm{of}\;\mathrm{flowers}\;\mathrm{per}\;\mathrm{plant}\;}\times 100$$

### 4.5. Essential Oil Analyses

The essential oil content of air-dried flowers was determined using Clavenger’s apparatus for determining oil quality in M_3_ and M_4_ selected mutants according to the method described in Santich [52]. Hydro distillation of 20 g of dried inflorescences takes four hours. The percentage of oil content was calculated as the average value of three measures. The essential oils were stored at 4 °C until analysis. Essential oil samples produced from M_4_ selected mutants (F/LF 5-2-1, F/HNOF 3-1-1, B/HNOF 4-3-1, B/HNO F 8-4-2 and M/HNOF 4-1-1) were subjected to GC-MS investigation to identify their oil composition. A percentage of the total chromatographic area was used to compute the relative content of each component. The compounds were identified based on a comparison of retention indices (RI) concerning n-alkanes (C7–C22), with relevant literature data and by matching their spectra with those of MS libraries (NIST 98, Willey, DuPage, USA) [53]. 

### 4.6. Statistical Analyses

The data recorded on different traits resulting from mutagens treatments were subjected to statistical analysis to find the individual and comparative effects of different mutagens. The mean, standard deviation and the LSD_0_._05_ were calculated using IBM SPSS statistics software version 22.

## 5. Conclusions

Chamomile is in high demand on the global market because of its vast medical uses and excellent pharmacological characteristics. Additionally, the usage of natural compounds rather than synthetic chemicals has increased. The present study was conducted to induce mutations suitable for mechanical harvesting, high flower productivity, oil content and oil quality in three German chamomile populations using sodium azide and gamma radiation treatments. Different morphological mutants of the three populations’ M_2_ population were isolated. These mutants involved traits affecting the flowers’ fresh weight, flowers’ dry weight, number of flowers, plant height, stem diameter, number of branches, days to flowering, oil content % and oil colors. Most of the useful mutations obtained in the M_3_ generation resulted from using gamma rays (11 mutants) instead of the 7 mutants obtained using sodium azide. Five promising mutants were selected in the M_4_ generation based on their characteristics, especially flower yield, oil content and oil quality. Two out of five mutants (F/LF5-2-1 and B/HNOF 8-4-2) could be integrated as potential parents into breeding programs for a high number of flowers, high oil content and oil quality traits. The selected desirable mutants will directly be used for commercial production after registration. 

## Figures and Tables

**Figure 1 plants-11-02940-f001:**
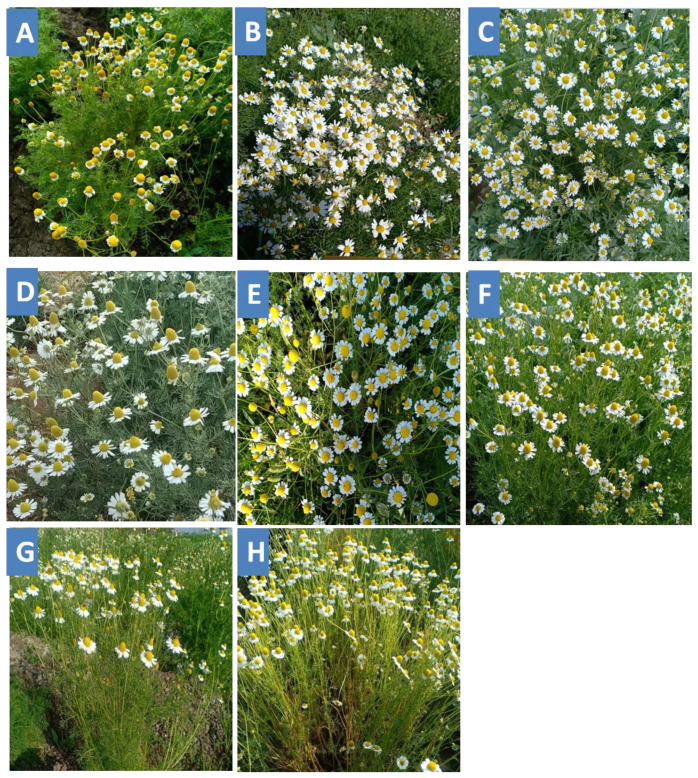
Phenotype of the selected five promising mutants and their control plants. (**A**). Control plant of Benysuef population. (**B**). BHNOF 4-3-1 mutant. (**C**). B/HNOF 8-4-2 mutant. (**D**). Control plant of Fayoum population. (**E**). F/HNOF 3-1-1 mutant. (**F**). F/LF5-2-1mutant. (**G**). Control plant of Menia population. (**H**). M/HNOF 4-1-1 mutant.

**Figure 2 plants-11-02940-f002:**
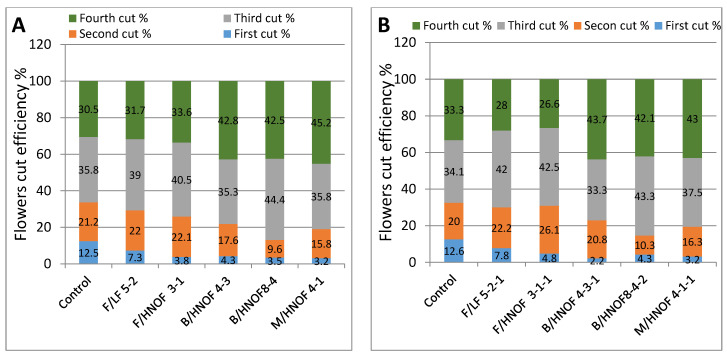
Flower cut efficiency percentage of selected promising mutants. (**A**). Flower cut efficiency (%) in M_3_ generation. (**B**). Flower cut efficiency (%) in M4 generation.

**Figure 3 plants-11-02940-f003:**
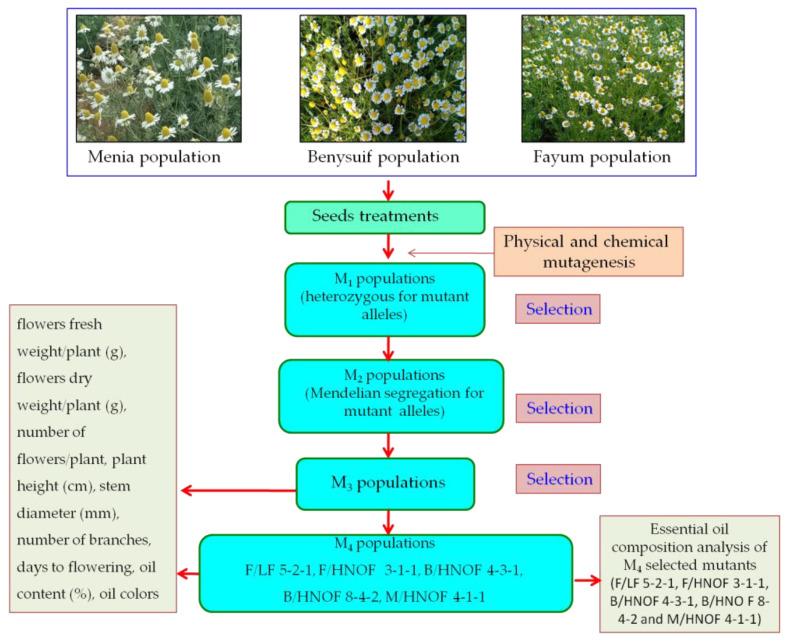
General procedures of the field experiments for the induction of promising mutations of three populations of German chamomile.

**Table 1 plants-11-02940-t001:** The number and kind of different mutants produced by gamma rays and sodium azide in M_2_ generation of the three German chamomile populations: Fayoum (Fa), Benysuif (Be), Menia (Me).

	Gamma Rays (Gy)												Sodium Azide (mol/mL)									
Mutation Type	100			200			300			400			0.001			0.002			0.003			Total
	Fa	Be	Me	Fa	Be	Me	Fa	Be	Me	Fa	Be	Me	Fa	Be	Me	Fa	Be	Me	Fa	Be	Me	
Dwarf	-	1	1	-	-	-	2	1	-		1	-	1	-	1	-	-	1	-	-	-	9
Semi-dwarf	-	-	1	-	2	1	1	1	-	1	-	1	1	1	-	1	-	1	1	-	-	13
Tall	3	2		1	1	1	-	1	-		-	-	1	-	1	-	-	2	1	-	1	15
Large stem diameter	1	1		1	-	1	-	-	-		-	-	-	-	-	1	-	1		1	1	8
High number of branches	2	2	1	1	-	1	1	-	-	-	1	1	-	-	1	1	-	-	1	3	-	16
Early flowering	1	3	-	1	4	-	1	-	1	1	-	2	1	-	1	-	1	-	1	1	1	20
Late flowering	1	2	1	1	-	2	3	-	-	1	1	-	-	1	-	1		2	2	-	-	18
High number of flowers	-	1	2	1	2	1	-	1	1	1	-	1	1	-	2	1	2	3	1	4	-	25
Total	8	12	6	6	9	7	8	4	2	4	3	5	5	2	6	5	3	10	7	9	3	124
	26			22			14			12			13			18			19			
	74												50									

**Table 2 plants-11-02940-t002:** Mean performance of morphological features, oil content and oil colors for selected M_3_ mutants of German chamomile populations.

	Mutants	Mutagens	Flowers Fresh Weight (g)	Flowers Dry Weight (g)	No. of Flowers	Plant Height (cm)	Stem Diameter (mm)	No. of Branches	Days to Flowering	Oil Content (%)	Oil Colors
**Fayoum**	Control	-	211.9 ^c^ ± 40.1	43.7 ^b^ ± 5.9	1089.7 ^b^ ± 319.6	76 ^b^ ± 7.9	10.7 ^bc^ ± 1.5	15.3 ^ab^ ± 2.5	123.3 ^a^ ± 24.6	0.91 ^b^ ± 0.64	Blue
	F/EF 5-1	SA 0.001	143.4 ^e^ ± 21.7	33.9 ^c^ ± 6.7	780 ^b^ ± 119.1	68.3 ^bc^ ± 12.6	8 ^c^ ± 3.6	16.3 ^a^ ± 1.5	113.3 ^a^ ± 5.8	0.89 ^b^ ± 0.2	Blue
	F/LF 2-1	G 200	168.7 ^d^ ± 24.4	38.3 ^c^ ± 6.1	606.6 ^d^ ± 71.4	61.3 ^c^ ± 9.1	11.3 ^ab^ ± 3.2	12.3 ^b^ ± 4.9	138 ^a^ ± 9.5	0.96 ^b^ ± 0.15	Blue
	**F/LF 5-2**	G 300	**571.3 ^a^** ± 220.6	**131.5 ^a^** ± **50.5**	1627.6 ^a^ ± 556.8	**97.6 ^a^** ± 6.8	**12 ^ab^** ± **1.7**	**17.6 ^a^** ± **2.1**	**136.3 ^a^** ± **3.2**	**1.77 ^a^** ± **0.46**	**Very light blue**
	F/LF 6-3	G 400	189.1 ^d^ ± 51.6	51.4 ^b^ ± 6.7	940.3 ^b^ ± 205.7	55.7 ^d^ ± 8.1	7 ^c^ ± 5.2	14.3 ^ab^ ± 4.6	130.3 ^a^ ± 2.1	1.01 ^b^ ± 0.3	Blue
	F/HNOF 3-1	SA 0.001	300.7 ^b^ ± 91.6	50.8 ^b^ ± 10.6	1113.6 ^c^ ± 82.5	67.6 ^c^ ± 5	12 ^a^ ± 3.1	12 ^ab^ ± 4.5	137.3 ^a^ ± 8.5	1.09 ^b^ ± 0.17	Blue
**Benysuef**	Control	-	212.3 ^e^ ± 32.6	48.2 ^e^ ± 9.1	1224 ^d^ ± 64.5	80.7 ^a^ ± 8.1	9.7 ^b^ ± 1.2	14 ^d^ ± 1	130 ^cd^ ± 18.1	0.89 ^a^ ± 0.21	Blue
	B/EF 3-1	G 100	276.5 ^d^ ± 41.3	65.3 ^d^ ± 7.4	966.7 ^g^ ± 341.9	71 ^b^ ± 1.7	15.3 ^a^ ± 2.5	17.7 ^cd^ ± 4.1	103.6 ^f^ ± 16.5	0.81 ^a^ ± 0.1	Blue
	B/LF 2-1	G 100	422.6 ^b^ ± 109.3	94.8 ^b^ ± 27.3	1619.3 ^c^ ± 45.4	69.7 ^b^ ± 13.8	10.7 ^ab^ ± 4.1	21.6 ^b^ ± 10.4	137.6 ^b^ ± 1.5	0.91 ^a^ ± 0.06	Blue
	B/HNOF 2-1	G 100	228 ^e^ ± 60.5	55.6 ^e^ ± 21.6	1178.7 ^e^ ± 49.6	72.3 ^b^ ± 10.7	11 ^b^ ± 2	21 ^cd^ ± 8.1	151 ^a^ ± 5.2	0.9 ^a^ ± 0.36	Blue
	B/HNOF 3-2	G 200	308.3 ^c^ ± 40.8	70.4 ^c^ ± 6.6	1097.3 ^f^ ± 46.6	67 ^b^ ± 9.6	9.7 ^b^ ± 5.5	20.3 ^c^ ± 9.4	113.3 ^d^ ± 8.6	0.79 ^a^ ± 0.16	Blue
	B/HNOF 4-3	G 200	410.7 ^b^ ± 201.7	94.3 ^b^ ± 46.6	1798.7 ^b^ ± 958.6	73.6 ^b^ ± 11.2	10 ^b^ ± 1.7	31.3 ^a^ ± 11.1	136 ^bc^ ± 5.5	1.09 ^a^ ± 0.15	Light brown
	**B/HNOF8-4**	**SA 0.003**	**533.4 ^a^** ± **229.3**	**116.6 ^a^** ± **55.1**	**2181 ^a^** ± **920.1**	**79.7 ^a^** ± **4.5**	**12 ^ab^** ± **4.3**	**22 ^bc^** ± **7**	**123.3 ^cd^** ± **3.1**	**1.29 ^a^** ± **0.25**	**Very light blue**
**Menia**	Control	-	188.9 ^g^ ± 8.2	44.4 ^d^ ± 4.5	1194.7 ^f^ ± 62.3	80 ^a^ ± 10	9.3 ^d^ ± 1.5	13 ^c^ ± 2.6	135.3 ^c^ ± 0.6	0.88 ^a^ ± 0.026	Blue
	M/EF 4-1	SA 0.001	536.6 ^a^ ± 57.7	114.1 ^a^ ± 12.7	2172 ^b^ ± 455.6	80 ^a^ ± 10	11 ^cd^ ± 1.7	21 ^b^ ± 3.6	94.3 ^e^ ± 5.8	0.96 ^a^ ± 0.47	Blue
	M/EF 5-2	SA 0.003	578 ^b^ ± 148.6	120 ^a^ ± 24.3	1832 ^c^ ± 431.8	87 ^a^ ± 11.5	16 ^bc^ ± 6.3	26 ^ab^ ± 7.8	96 ^e^ ± 7.5	0.98 ^a^ ± 0.225	Blue
	M/LF 3-1	G 200	320 ^f^ ± 46.7	75 ^c^ ± 10.1	1196 ^g^ ± 268.7	73 ^a^ ± 13.3	13 ^ab^ ± 4.3	21 ^bc^ ± 2.5	144 ^ab^ ± 10.8	0.95 ^a^ ± 0.37	Blue
	M/LF 5-2	G 100	401 ^d^ ± 44.8	108 ^a^ ± 5.9	1328 ^d^ ± 554.1	82 ^a^ ± 7.5	17 ^a^ ± 2.9	30 ^ab^ ± 5.3	150 ^a^ ± 3.1	1 ^a^ ± 0.1	Blue
	**M/HNOF 4-1**	**G 400**	**571.7 ^a^** ± **126.7**	**124.5 ^a^** ± **30.7**	**2149 ^a^** ± **990.8**	**80.3 ^a^** ± **4.1**	**11 ^cd^** ± **1.7**	**19 ^b^** ± **6.1**	**140.3 ^bc^** ± **5.5**	1.09 ^a^ ± 0.2	**Very dark yellow**
	M/HNOF 6-2	SA 0.001	349.5 ^e^ ± 74	73.3 ^c^ ± 16.3	1246.3 ^fg^ ± 362.1	81.7 ^a^ ± 7.6	15.3 ^a^ ± 3.2	29.3 ^a^ ± 4.9	100 ^e^ ± 2.2	0.96 ^a^ ± 0.31	Blue
	M/HNOF 7-3	SA 0.002	451.4 ^c^ ± 20.3	95.1 ^b^ ± 7.7	1622.3 ^e^ ± 33.5	59 ^a^ ± 41.4	12 ^c^ ± 1	20.7 ^b^ ± 1.5	124 ^d^ ± 13.5	0.97 ^a^ ± 0.35	Blue

Notes: Values (mean ± SE) with different letters in the same column are significantly different at *p* < 0.05 and vice versa. Different letters in the same column refer to the significant difference among genotypes at *p* < 0.05. Green color refers to the two promising mutants that could be integrated as potential parents into breeding programs.

**Table 3 plants-11-02940-t003:** Mean performance of morphological features, oil content and oil colors for the selected five M_4_ promising mutants of German chamomile populations.

Mutants	Mutagens	Flowers Fresh Weight (g)	Flowers Dry Weight (g)	No. of Flowers	Plant Height (cm)	Stem Diameter (mm)	No. of Branches	Days to Flowering	Oil Content (%)	Oil Colors
**Control**	-	188.7 **^ab^** ± 61.6	47.3 **^cd^** ± 13.9	1079.1 **^d^** ± 240.8	77.2 **^c^** ± 2.6	9.3 **^b^** ± 1.15	13.6 **^c^** ± 2.1	126.5 **^b^** ± 11.1	0.9 **^c^** ± 0.24	Blue
**F/LF 5-2-1**	G 300	505.1 **^a^** ± 262.7	115.5 **^b^** ± 47.6	1595.3 **^b^** ± 523.2	82.4 **^a^** ± 2.6	14 **^a^** ± 2.1	16.2 **^bc^** ± 2	138.1 **^a^** ± 3.1	1.75 **^a^** ± 0.11	Light blue
F/HNOF 3-1-1	SA 0.001	206 **^b^** ± 48.1	46.6 **^d^** ± 6.6	1015.4 **^e^** ± 46.7	71.8 **^c^** ± 9.6	117 **^ab^** ± 5.5	11.5 **^c^** ± 9.4	131.2 **^a^** ± 8.6	0.99 **^bc^** ± 0.019	Blue
B/HNOF 4-3-1	G 200	314.1 **^ab^** ± 115.4	72.4 **^bc^** ± 33.5	1380.5 **^c^** ± 315.1	75 **^bc^** ± 7.2	10.6 **^b^** ± 2.1	25.4 **^a^** ± 6.4	136 **^a^** ± 3.2	1.03 **^bc^** ± 0.19	Light brown
**B/HNOF 8-4-2**	SA 0.003	510.2 **^a^** ± 192.1	111.5 **^a^** ± 46.7	1900 **^a^** ± 549	77.7 **^ab^** ± 3	137 **^ab^** ± 3.2	20 **^ab^** ± 4.1	124.4 **^b^** ± 1.5	1.22 **^b^** ± 0.18	Very light blue
M/HNOF 4-1-1	G 400	433.9 **^ab^** ± 177.5	95.6 **^bc^** ± 48.3	1930 **^a^** ± 931.6	79.3 **^ab^** ± 2	127 **^ab^** ± 1.5	20 **^ab^** ± 2.1	138.5 **^a^** ± 4.5	0.99 **^bc^** ± 0.067	Very dark yellow

Notes: Values (mean ± SE) with different letters in the same column are significantly different at *p* < 0.05 and vice versa. Different letters in the same column refer to the significant difference among genotypes at *p* < 0.05. Green color refers to the two promising mutants that could be integrated as potential parents into breeding programs.

**Table 4 plants-11-02940-t004:** GC-MS essential oil composition of the selected five promising M_4_ mutants.

	COMPOUND	LRI	RT	Basic Composition of Essential Oil (%)					
				Control	F/LF 5-2-1	F/HNOF 3-1-1	B/HNOF 4-3-1	B/HNO F 8-4-2	M/HNOF 4-1-1
				Peak Area %					
**1**	**2,4,5-Trimethyl-1,3-dioxolane**	**695.4**	**3.236**	**-**	**-**	**-**	**-**	-	0.78
2	Amyl ethyl ether	709	3.282	-	-	-	-	-	0.77
3	Trimethylsilylmethanol	493.1	4.197	2.73	4.52	2.37	6.13	4.52	32.4
4	Acetone, dimethyl acetyl	577.6	4.277	-	-	-	0.22	0.17	1.06
5	Ethyl α-methylbutyrate	405.2	4.821	0.56	1.67	0.53	0.19	0.77	0.99
6	Propyl 2-methylbutanoate	741.5	7.184	0.34	0.6	0.28	-	-	-
7	3,6-Heptadien-2-ol, 2,5,5-trimethyl-, (E)-	996.9	8.769	0.4	-	0.28	-	0.24	0.95
8	p-Cymene	959.3	9.593	1.41	1.1	0.73	0.45	1.74	-
9	Benzeneacetaldehyde	767.8	10.205	-	0.76	-	-	-	-
10	γ-Terpene	1018.7	10.714	-	-	-	-	0.41	-
11	Artemisia ketone	1026.8	10.795	1.82	2.17	1.85	1.62	1.27	0.77
12	Artemisia alcohol (2,5,5-trimethylhepta-2,6-dien-4-ol)	1300	11.527	0.48	-	0.4	-	0.34	-
13	Mequinol	572.2	11.722	-	0.75	-	-	-	-
14	(-)-Borneol	935.4	14.354	0.88	1.27	0.93	-	0.88	-
15	cis-3-Hexenyl-α-methylbutyrate	1174.5	16.745	-	-	-	-	0.4	-
16	trans-2-Hexenyl valerate	1302	17.02	-	-	-	-	0.37	-
17	4,8-Dimethylnona-3,8-dien-2-one	1401	18.01	-	-	-	-	0.31	-
18	n-Decanoic acid	1618.6	21.186	-	1.84	3.37	1.26	0.86	-
19	cis-β-Farnesene	2393.2	23.932	2.44	-	-	-	-	-
20	(E)-β-Farnesene	2393.8	23.938	-	3.47	2.23	2.04	1.74	-
21	Dehydrosesquicineole	2431.6	24.316	0.8	-	-	-	0.33	-
22	γ-Cadinene	2571.2	25.712	-	0.68	-	-	-	-
23	Spathulenol	2762.3	27.623	2.9	2.41	0.97	6.05	4.9	-
24	Aristolene epoxide	1419.2	28.081	-	-	-	0.41	0.39	-
25	4,4,7a-Trimethyl-2,3,3a,4,5,7a-hexahydro-1H-inden-1-one	1185	28.739	0.81	0.81	0.54	2.41	1.24	-
26	Aromadendrene oxide-(1)	1510.2	28.991	-	-	-	0.49	-	-
27	6-[1-(Hydroxymethyl)vinyl]-4,8a-dimethyl-1,2,4a,5,6,7,8,8a-octahydro-2-naphthalenol	1521.6	29.105	-	-	0.36	1.58	0.77	-
28	α-epi-Cadinol	1542.8	29.317	-	4.84	-	0.79	0.49	-
29	Bisabolol oxide B	1517.6	29.889	10.71	6.87	6.67	20.62	5.85	1.27
30	β-Santalol	1529.6	30.009	3.19	3.22	3.55	3.27	1.99	8.32
31	Costol	1553.1	30.244	-	-	-	0.48	-	-
32	Bisabolone oxide A	1497.1	30.684	12.36	10.2	7.92	4.23	6.93	1.34
33	Calarene epoxide	1595.4	30.667	-	-	-	-	0.41	-
34	α-Bisabolone oxide A	1316.3	30.713	2.75	-	-	-	-	-
35	Chamazulene	1431.3	31.863	1.58	1.35	2.74	0.62	13.93	-
36	Alpha-bisabolol	1275.6	32.59	36.34	41.11	45.91	33.19	39.52	47.32
37	Campherene-2,13-diol	1422.1	34.055	-	-	0.3	-	-	-
38	cis-ene-yne-Dicycloether	1375.4	35.834	7.86	9.25	9.57	10.06	7.39	2.26
39	trans ene-yne-Dicycloether	1400	36.08	0.98	1.11	1.25	1.21	0.74	0.75
40	(E)-2-(Hepta-2,4-diyn-1-ylidene)-1,6-dioxaspiro[4.4]non-3-ene	1640.2	37.482	0.49	-	0.58	0.54	-	-
41	Palmitic acid	1431.8	37.654	1.63	-	2.65	0.44	0.67	-
42	Linoleic acid	2024.9	41.585	-	-	0.47	-	-	-
43	Oleic Acid	2037.5	41.711	-	-	0.44	-	-	-
44	(Z)-18-Octadec-9-enolide	2061.5	41.951	0.59	-	0.62	-	-	-
45	Tricosane	2885.4	45.19	-	-	0.52	-	-	-
46	Triacontane	3968.3	49.019	1.41	-	1.55	0.76	0.46	-

RT is Retention Time and LRI is Linear Retention Indices. Green color refers to the two promising mutants that could be integrated as potential parents into breeding programs.

## Data Availability

The data presented in this study are available upon request from the corresponding authors.

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
