# Peer review of "Improvement of German Chamomile (Matricaria recutita L.) for Mechanical Harvesting, High Flower Yield and Essential Oil Content Using Physical and Chemical Mutagenesis"

_plants, 2022, doi:10.3390/plants11212940_

Round 1
Reviewer 1 Report
This paper describes the screening of useful mutant lines of German chamomile using mutation techniques. The promising mutant lines seem very useful for essential oil production, and the stability of mutant phenotype is well confirmed in M3 and M4 generation. Particularly, mutant screening suitable for mechanical harvesting is interesting to me. In general, the manuscript is well written, however the following points need to be considered before publication.
Line 103: The three populations (Fa, Be and Me) should be defined here. Also, I suggest to describe the reason why the authors chose those three populations.
Line 107-111: I think the content of this paragraph is redundant and unnecessary.
Line 113: I suggest to describe the method for mutagen treatment briefly, because the Materials & Methods section is placed after Discussion section. Also, describe the survival rate of the treated seeds, so that the readers can understand the strength of mutagen treatment.
Line 123: The population size of M2 (Total number of plants) should be described in this section, otherwise the readers cannot understand the absolute value of the mutation frequency.
Line 132: “the fourth mutant” should be specified by the mutant ID.
Line 137: It seems NO data is shown for “cut efficiency percentage” of M2 plants.
Table 2, 3 and 4: Indicate the meaning of green color in a footnote. Indicate the meaning of superscript letters in Table 2. Not only doses but also the types of mutagen should be shown in Table 2 and 3 (for example, G100 and SA0.001). Define the abbreviation of “LRT” and “RT” in a footnote of Table 4.
Figure 1B: “d” of the “Second cut” is missing.
Line 219-222: Meaning is unclear to me. “high sensitivity” of what? What is the difference between “genetic instability” and mutation? “individual changes” of what? Could you cite some paper regarding the fact that the individual changes are frequently tolerated by a well-organized co-adapted gene complex?
Line 243: What is the “minor deficits”?
Author Response
Response to Reviewer 1 Comments
Manuscript ID: plants-1969401
Manuscript Title: Promising mutations for mechanical harvesting, high flowers yield and essential oil content in German chamomile (Matricaria recutita L.) using physical and chemical mutagenesis
We are very glad that Reviewer 1 highly evaluated our manuscript, and provided constructive comments and valuable suggestions that have helped us further improvement of the quality of our manuscript. All the changes made in response to the Reviewer 1 comments were corrected as track changes in the revised manuscript. We have addressed all of your queries and improved our manuscript following your suggestions as you can see in our point-to-point responses to your comments below.
Comments and Suggestions for Authors
This paper describes the screening of useful mutant lines of German chamomile using mutation techniques. The promising mutant lines seem very useful for essential oil production, and the stability of mutant phenotype is well confirmed in M3 and M4 generation. Particularly, mutant screening suitable for mechanical harvesting is interesting to me. In general, the manuscript is well written, however the following points need to be considered before publication.
Line 103: The three populations (Fa, Be and Me) should be defined here. Also, I suggest to describe the reason why the authors chose those three populations.
Response: Done accordingly
Line 107-111: I think the content of this paragraph is redundant and unnecessary.
Response: Done accordingly
Line 113: I suggest to describe the method for mutagen treatment briefly, because the Materials & Methods section is placed after Discussion section. Also, describe the survival rate of the treated seeds, so that the readers can understand the strength of mutagen treatment.
Response: Done accordingly
Line 123: The population size of M2 (Total number of plants) should be described in this section, otherwise the readers cannot understand the absolute value of the mutation frequency.
Response: Done accordingly (a great note. Many thanks)
Line 132: “the fourth mutant” should be specified by the mutant ID.
Response: Done accordingly
Line 137: It seems NO data is shown for “cut efficiency percentage” of M2 plants.
Response: Done accordingly Table S2 was added
Table 2, 3 and 4: Indicate the meaning of green color in a footnote. Indicate the meaning of superscript letters in Table 2. Not only doses but also the types of mutagen should be shown in Table 2 and 3 (for example, G100 and SA0.001).
Response: Done accordingly
Define the abbreviation of “LRT” and “RT” in a footnote of Table 4.
Response: Done accordingly
Figure 1B: “d” of the “Second cut” is missing.
Response: Done accordingly
Line 219-222: Meaning is unclear to me. “high sensitivity” of what? What is the difference between “genetic instability” and mutation? “individual changes” of what? Could you cite some paper regarding the fact that the individual changes are frequently tolerated by a well-organized co-adapted gene complex?
Response: Done accordingly (Thanks to the reviewer for his/her care about the quality of the article- this part is deleted)
Line 243: What is the “minor deficits”?
Response: Done accordingly (minor effects or minor alterations) deficits is a mistake it was corrected.

Reviewer 2 Report
Dear author(s),
The following points should clearly be corrected and addressed for readers:
Title
1. L2-4, short title will be better than the present title of manuscript (mn).
1. “Improvement of German chamomile (Matricaria recutita L.) using mutagenesis”, or
2. “Improvement of German chamomile (Matricaria recutita L.) for mechanical harvesting, high flowers yield and essential oil content using mutagenesis”.
2. As an alternative suggestion, at least, change “mutations” to “mutants” in title.
Abstract
3. L27, all kind of researchers will read your mn and some of them have not enough knowledge about chamomile. Thus, you should give more information for readers on usage of the plant. Which organs are used? Flowers, leaves, seeds, or whole plant??? What is the importance of the study? The first sentence in “Abstract” should be explained these questions?
Flowers of German chamomile (Matricaria recutita L.), one of the most important medicinal plants, is used for various applications. Essential improvement goals of chamomile are considered to be high flowers yield and oil content, and suitable for mechanical harvest as well. Aims of the present study were improved for flower yield, oil content and mechanical harvestable of German chamomile via chemical and physical mutagens.
4. L32, remove “GC-MS analysis method was used to identify the oil composition of the promising mutants.” Readers can see details of oil detection protocol in M&M.
5. L34 and 37, M3 and M4 should be written properly (as M3 and M4). That is numbers should be subcritical written.
Keywords
6. L47, keywords could be ordered for their importance or alphabetically. Scientific name could be written as Matricaria recutita.
Introduction
7. L62, Scientific name (Matricaria recutita L.) could be written after German chamomile.
8. L69, a new paragraph should be used after “Unfortunately, ….”
9. L93, check sentence.
Results
10. L107, delete the first sentence. You have already mentioned aim of the study.
11. L109, please be aware of mutations and mutants. As you know, mutation is created by mutagens but mutant is an individual plant as a result of mutation. Change mutations to mutants.
12. L112, subtitles must be compatible. M2 populations but M3 and M4 mutants.
13. L113, I do not understand this sentence. Did you mean “Various morphological mutations were found in three M2 populations?” Please use clear and understandable sentences.
14. Please read “Spectrum and frequency of induced mutations in chickpea”. http://oar.icrisat.org/1200/
15. L12, a blank before high
16. L124, change have to had
17. L127, change during to in and M2 to M2
18. L129, change M2 to M2
19. L129, correct flowers’… as fresh weight of flowers per plant. Others, too.
20. L138, change M2 to M2
21. L141, I strongly suggest that passive sentence should be preferred.
22. L146, L129, change M3 to M3
23. L156, change very important to desirable
24. L161, change M3 to M3
25. L170, change M3 to M3
26. In Table 2, delete “plant”. You have already detailed traits studied in M&M.
27. In Table 2, change Number of flowers/branches to No flowers, No branches
28. In Table 2, use bracket as (%).
29. In Table 2, LSD can be removed since you gave SE or SD. Please explain them SE or SD.
30. L174, change M4 to M4
31. L175, change M3 to M3
32. L180, change theresults to The results
33. L186, change % to (%)
34. In Fig 1, change % to (%)
35. L189-190, change M3/M4 to M3/M4
36. L194, L170, change M4 to M4
37. In Table 3, please consider my suggestions in Table 2.
38. In Table 3, letters, LSD and SE. One of them is enough for readers.
39. L199, change M4 to M4
40. L205, a blank before a new sentence.
41. L224, blank after words
42. L228, change figure 2 to Figure 2.
43. In Fig 2, photos could be close up.
44. L234, change M4 to M4
45.
Discussion
46. L245-246, delete g, cm, % and mm
47. L247, change M2 to M2
48. L258, change M2 to M2
49. L259-260, et al. and et al. were not written as italic.
50. L260, change Mungo to mungo. Write common name of plants.
51. L263, Vigna should be italicized.
52. L276-281, Blanks between words.
53. L281, correct Lal, to Lal
M&M
54. L301, change Plants to plant.
55. L305, please provide longitude, latitude and altitude of the location.
56. L307, change withgamma to with gamma.
57. L308 provide chemical formula of sodium azide as (NaNO3).
58. L309, change “Gamma radiation” to Physical mutagen”
59. L311, CO60 or Co60, change Co60 to Co60
60. L313, change “Chemical mutagen: sodium azide (SA)” to “Chemical mutagen”
61. L318, correct subtitle as Agronomic practices and data collection
62. L319, a blank before the irradiated, ….
63. L321-323, it is meaningless to sow and grew of M1 and M2 generations in the replicated trials. M1 and M2 are similar to F1 (genetically heterozygous but phenotypically homogeny) and F2 (segregations). Your results will not be affected due to unnecessary experiments. Please delete or remove this sentence and correct the related sentences in the whole text.
64. L324-325, readers want to see irrigation quantity and irrigation system. Following paper (https://doi.org/10.3390/ agronomy12030557) can be read for detailed information. If it is suitable, it can be used as a reference.
65. L325, routine managements should be explained. Weed control, fertilization etc.
66. L327, replication is unnecessary, please delete again….
67. L336, which generation was data collected in?
68. L328, change M1 to M1
69. L330, the selected M2 plants in M3 can be grown in RCBD with replication. It is possible for later generations, too.
70. L335-336, delete M2 and correct the sentence.
71. L337, change “flowers’ weight of flowers/plant” to fresh weight of flowers per plant. Please correct others.
72. L339, change % to (%)
73. L346, correct subtitle as Essential oil content or Essential oil analyses since analysis is more than one.
74. L347-349, check the sentence and please write clearly.
75. L348, change M3 to M3
76. L352, M4 selected mutants??? What does it mean? Did you continue selection in M4? Or Did you select mutants in M2 and then sow mutants as M4? Please explain this important detail in here and Diagram 1.
77. L357, delete : and change analysis to analyses because there was more than one analysis.
78. L358, change have been to were
Diagram
79. M0, it means mutant or mutated zero. This sound meaningless. Irradiated seeds is better. Selection should be started in M2 and continued in M3. Change % to (%).
Conclusions
80. L372, change M2 to M2
81. L373-374, delete g, cm, % and mm
82. L375, change M3 to M3
83. L247, change M2 to M2
84. L277, change M4 to M4
85. L380, A conclusion sentence should be written for readers about mutation breeding on chamomile.

Author Response
Response to Reviewer 2 Comments
Manuscript ID: plants-1969401
Manuscript Title: Promising mutations for mechanical harvesting, high flowers yield and essential oil content in German chamomile (Matricaria recutita L.) using physical and chemical mutagenesis
We are very thankful for the great scientist (Reviewer 2) who highly evaluated our manuscript, and provided constructive comments and valuable suggestions that have helped us further improvement of the quality of our manuscript. All the changes made in response to the Reviewer 1 comments were corrected as track changes in the revised manuscript. We have addressed all of your queries and improved our manuscript following your suggestions as you can see in our point-to-point responses to your comments below.
Dear author(s),
The following points should clearly be corrected and addressed for readers:
Title
- L2-4, short title will be better than the present title of manuscript (mn).
- “Improvement of German chamomile (MatricariarecutitaL.) using mutagenesis”, or
- “Improvement of German chamomile (MatricariarecutitaL.) for mechanical harvesting, high flowers yield and essential oil content using mutagenesis”.
- As an alternative suggestion, at least, change “mutations” to “mutants” in title.
Response: Done accordingly
Abstract
- L27, all kind of researchers will read your mn and some of them have not enough knowledge about chamomile. Thus, you should give more information for readers on usage of the plant. Which organs are used? Flowers, leaves, seeds, or whole plant??? What is the importance of the study? The first sentence in “Abstract” should be explained these questions?
Flowers of German chamomile (MatricariarecutitaL.), one of the most important medicinal plants, is used for various applications. Essential improvement goals of chamomile are considered to be high flowers yield and oil content, and suitable for mechanical harvest as well. Aims of the present study wereimproved for flower yield, oil content and mechanical harvestable of German chamomile via chemical and physical mutagens.
Response: Done accordingly (thank you so much for positive suggestions)
- L32, remove “GC-MS analysis method was used to identify the oil composition of the promising mutants.” Readers can see details of oil detection protocol in M&M.
Response: Done accordingly
- L34 and 37, M3 and M4 should be written properly (as M3 and M4). That is numbers should be subcritical written.
Response: Done accordingly
Keywords
- L47, keywords could be ordered for their importance or alphabetically.Scientific name could be written as Matricariarecutita.
Response: Done accordingly
Introduction
- L62, Scientific name (Matricariarecutita) could be written after German chamomile.
Response: Done accordingly
- L69, a new paragraph should be used after “Unfortunately, ….”
Response: Done accordingly
- L93, check sentence.
Response: Done accordingly
Results
- L107, delete the first sentence. You have already mentioned aim of the study.
Response: Done accordingly
- L109, please be aware of mutations and mutants. As you know, mutation is created by mutagens but mutant is an individual plant as a result of mutation. Change mutations to mutants.
- L112, subtitles must be compatible. M2 populations but M3 and M4 mutants.
- L113, I do not understand this sentence. Did you mean “Various morphological mutations were found in three M2 populations?”Please use clear and understandable sentences.
Response: Done accordingly
- Please read “Spectrum and frequency of induced mutations in chickpea”. http://oar.icrisat.org/1200/
Response: Done accordingly
- L12, a blank before high
Response: Done accordingly
- L124, change have to had
Response: Done accordingly
- L127, change during to in and M2 to M2
Response: Done accordingly
- L129, change M2 to M2
Response: Done accordingly
- L129, correct flowers’… as fresh weight of flowers per plant. Others, too.
Response: Done accordingly
- L138, change M2 to M2
Response: Done accordingly
- L141, I strongly suggest that passive sentence should be preferred.
Response: Done accordingly
- L146, L129, change M3 to M3
Response: Done accordingly
- L156, change very important to desirable
Response: Done accordingly
- L161, change M3 to M3
Response: Done accordingly
- L170, change M3 to M3
Response: Done accordingly
- In Table 2, delete “plant”. You have already detailed traits studied in M&M.
Response: Done accordingly
- In Table 2, change Number of flowers/branches to No flowers, No branches
Response: Done accordingly
- In Table 2, use bracket as (%).
Response: Done accordingly
- In Table 2, LSD can be removed since you gave SE or SD. Please explain them SE or SD.
Response: Done accordingly
- L174, change M4 to M4
Response: Done accordingly
- L175, change M3 to M3
Response: Done accordingly
- L180, change theresults to The results
Response: Done accordingly
- L186, change % to (%)
Response: Done accordingly
- In Fig 1, change % to (%)
Response: Done accordingly
- L189-190, change M3/M4 to M3/M4
Response: Done accordingly
- L194, L170, change M4 to M4
Response: Done accordingly
- In Table 3, please consider my suggestions in Table 2.
Response: Done accordingly
- In Table 3, letters, LSD and SE. One of them is enough for readers.
Response: Done accordingly
- L199, change M4 to M4
Response: Done accordingly
- L205, a blank before a new sentence.
Response: Done accordingly
- L224, blank after words
Response: Done accordingly
- L228, change figure 2 to Figure 2.
Response: Done accordingly
- In Fig 2, photos could be close up.
Response: Done accordingly
- L234, change M4 to M4
Response: Done accordingly
Discussion
- L245-246, delete g, cm, % and mm
Response: Done accordingly
- L247, change M2 to M2
Response: Done accordingly
- L258, change M2 to M2
Response: Done accordingly
- L259-260, et al. and et al. were not written as italic.
Response: Done accordingly
- L260, change Mungo to mungo. Write common name of plants.
Response: Done accordingly
- L263, Vigna should be italicized.
Response: Done accordingly
- L276-281, Blanks between words.
Response: Done accordingly
- L281, correct Lal, to Lal
Response: Done accordingly
M&M
- L301, change Plants to plant.
Response: Done accordingly
- L305, please provide longitude, latitude and altitude of the location.
Response: Done accordingly
- L307, change withgamma to with gamma.
Response: Done accordingly
- L308 provide chemical formula of sodium azide as (NaNO3).
Response: Done accordingly
- L309, change “Gamma radiation” to Physical mutagen”
Response: Done accordingly
- L311, CO60 or Co60, change Co60 to Co60
Response: Done accordingly
- L313, change “Chemical mutagen: sodium azide (SA)” to “Chemical mutagen”
Response: Done accordingly
- L318, correct subtitle as Agronomic practices and data collection
Response: Done accordingly
- L319, a blank before the irradiated, ….
Response: Done accordingly
- L321-323, it is meaningless to sow and grew of M1 and M2 generations in the replicated trials. M1 and M2 are similar to F1 (genetically heterozygous but phenotypically homogeny) and F2 (segregations). Your results will not be affected due to unnecessary experiments. Please delete or remove this sentence and correct the related sentences in the whole text.
Response: Done accordingly
- L324-325, readers want to see irrigation quantity and irrigation system. Following paper (https://doi.org/10.3390/ agronomy12030557) can be read for detailed information.If it is suitable, it can be used as a reference.
Response: Done accordingly
- L325, routine managements should be explained. Weed control, fertilization etc.
Response: Done accordingly
- L327, replication is unnecessary, please delete again….
Response: Done accordingly
- L336, which generation was data collected in?
Response: Done accordingly
- L328, change M1 to M1
Response: Done accordingly
- L330, the selected M2 plants in M3 can be grown in RCBD with replication. It is possible for later generations, too.
Response: Done accordingly
- L335-336, delete M2 and correct the sentence.
Response: Done accordingly
- L337, change “flowers’ weight of flowers/plant” to fresh weight of flowers per plant. Please correct others.
Response: Done accordingly
- L339, change % to (%)
Response: Done accordingly
- L346, correct subtitle as Essential oil content or Essential oil analyses since analysis is more than one.
Response: Done accordingly
- L347-349, check the sentence and please write clearly.
Response: Done accordingly
- L348, change M3 to M3
Response: Done accordingly
- L352, M4 selected mutants??? What does it mean? Did you continue selection in M4? Or Did you select mutants in M2 and then sow mutants as M4? Please explain this important detail in here and Diagram 1.
Response: Done accordingly
- L357, delete : and change analysis to analyses because there was more than one analysis.
Response: Done accordingly
- L358, change have been to were
Response: Done accordingly
Diagram
- M0, it means mutant or mutated zero. This sound meaningless. Irradiated seeds is better. Selection should be started in M2 and continued in M3. Change % to (%).
Response: Done accordingly
Conclusions
- L372, change M2 to M2
Response: Done accordingly
- L373-374, delete g, cm, % and mm
Response: Done accordingly
- L375, change M3 to M3
Response: Done accordingly
- L247, change M2 to M2
Response: Done accordingly
- L277, change M4 to M4
Response: Done accordingly
- L380, A conclusion sentence should be written for readers about mutation breeding on chamomile.

Reviewer 3 Report
The authors present results from their study of the effects of chemical and physical mutagens on the morphological and chemical features of the German chamomile Matricaria recutita L.
In my opinion, the aforementioned paper requires substantial improvements, with regard to the quality of presentation and English language. Below, I present more detailed comments.
1. Throughout the entire text I found numerous typographical and linguistical mistakes. Taking into account their number, I will indicate here only a few:
a. Page 2, line 93: “chamomile considers” – should be “chamomile is considered”
b. Page 2, line “harvesting completely manual four times due to it”. Wrong grammar
c. Page 3, lines 113 – 114 “Three populations of German chamomile were found to have M2 populations”. I do not understand. Furtheremore throughout the text the authors write alternatively “M2” or “M2”; “M3” or “M3” etc. Please use consistent naming.
d. In many places spaces are missing between words e.g. page 3, line 120 “ahigh”.
e. Page 4, line 161: “mutants showed stability”. What do you mean?
f. Page 4, line 263. Latin names should be given.
2. Regarding the scientific value of the text, my main concern is description of field experiments, which I found difficult to understand. I refer here to the following part: “The M1 plants were harvested separately in each treatment and controlled for obtaining seeds (What do you mean by controlled for obtaining seeds?). During the winter of 2020, three replications of each normal-appearing M1 plant from each treatment and its corresponding control (wild type) in each population.” So, do you mean that you did not collect mutants from the M1 population? If so, why? I simply do not understand this approach.
Author Response
Response to Reviewer 3 Comments
Manuscript ID: plants-1969401
Manuscript Title: Promising mutations for mechanical harvesting, high flowers yield and essential oil content in German chamomile (Matricaria recutita L.) using physical and chemical mutagenesis
We are very glad that Reviewer 3 highly evaluated our manuscript, and provided constructive comments and valuable suggestions that have helped us further improvement of the quality of our manuscript. All the changes made in response to the Reviewer 1 comments were corrected as track changes in the revised manuscript. We have addressed all of your queries and improved our manuscript following your suggestions as you can see in our point-to-point responses to your comments below.
Comments and Suggestions for Authors
The authors present results from their study of the effects of chemical and physical mutagens on the morphological and chemical features of the German chamomile Matricaria recutita L.
In my opinion, the aforementioned paper requires substantial improvements, with regard to the quality of presentation and English language. Below, I present more detailed comments.
- Throughout the entire text I found numerous typographical and linguistical mistakes. Taking into account their number, I will indicate here only a few:
- Page 2, line 93: “chamomile considers” – should be “chamomile is considered”
Response: Done accordingly
- Page 2, line “harvesting completely manual four times due to it”. Wrong grammar
Response: Done accordingly
- Page 3, lines 113 – 114 “Three populations of German chamomile were found to have M2 populations”. I do not understand.
Response: Done accordingly
Furtheremore throughout the text the authors write alternatively “M2” or “M2”; “M3” or “M3” etc. Please use consistent naming.
Response: Done accordingly
- In many places spaces are missing between words e.g. page 3, line 120 “ahigh”.
Response: Done accordingly
- Page 4, line 161: “mutants showed stability”. What do you mean?
Response: Done accordingly (corrected to performance stability)
- Page 4, line 263. Latin names should be given.
Response: Done accordingly
- Regarding the scientific value of the text, my main concern is description of field experiments, which I found difficult to understand. I refer here to the following part: “The M1 plants were harvested separately in each treatment and controlled for obtaining seeds (What do you mean by controlled for obtaining seeds?).
Response: Done accordingly (corrected)
During the winter of 2020, three replications of each normal-appearing M1 plant from each treatment and its corresponding control (wild type) in each population.” So, do you mean that you did not collect mutants from the M1 population? If so, why? I simply do not understand this approach.
Response: Normally the selection of mutations begins from M2 to avoid selecting of morphological mutations occurred due to some physiological changes. The phenotypes appeared due to environmental variation or seedling physiology are disappeared in M2

Reviewer 4 Report
Keywords: Arrange keywords in alphabetical order.
Methodology: What is the population size of 3 German chamomile populations? Write them.
Diagram 1: How do chimeric plants in M1 generation convert to non-chimeric in M2? Are there any cytological studies conducted to confirm mutation level in cell? Describe it.
Line 328: What is meant by a normal appearing plant in M1? According to my perception, mutant plants never appear normal just like untreated plants. Clarify the statement.
329: Why were control plants attributed as wild? Are populations of chamomiles growing as wild in different regions of Egypt? If they are domesticated then the word wild seems to be out of scope. Use untreated or control in place of wild.
Results: Mutation induction % of both physical and chemical mutations must be included in results for further studies about the selection of mutagen and its specific concentration.
Where is characterization of M1 generation? It is logical to begin mutation selection from 1st generation instead of 2nd generation. Explain it with scientific reasoning.
How homogeneity is maintained in each selected mutant within each selection as it is already mentioned in the introduction that is frequently outcrossed by insects.
Table 2: It is better to write the mutagen name in front of concentration to avoid confusion.
Author Response
Response to Reviewer 4 Comments
Manuscript ID: plants-1969401
Manuscript Title: Promising mutations for mechanical harvesting, high flowers yield and essential oil content in German chamomile (Matricaria recutita L.) using physical and chemical mutagenesis
We are very glad that Reviewer 4 highly evaluated our manuscript, and provided constructive comments and valuable suggestions that have helped us further improvement of the quality of our manuscript. All the changes made in response to the Reviewer 1 comments were corrected as track changes in the revised manuscript. We have addressed all of your queries and improved our manuscript following your suggestions as you can see in our point-to-point responses to your comments below.
Comments and Suggestions for Authors
Keywords: Arrange keywords in alphabetical order.
Response: Done accordingly
Methodology: What is the population size of 3 German chamomile populations? Write them.
Response: Done accordingly
Diagram 1: How do chimeric plants in M1 generation convert to non-chimeric in M2? Are there any cytological studies conducted to confirm mutation level in cell? Describe it.
Response: Done accordingly (corrected)
Line 328: What is meant by a normal appearing plant in M1? According to my perception, mutant plants never appear normal just like untreated plants. Clarify the statement.
Response: Done accordingly (corrected)
329: Why were control plants attributed as wild? Are populations of chamomiles growing as wild in different regions of Egypt? If they are domesticated then the word wild seems to be out of scope. Use untreated or control in place of wild.
Response: Done accordingly (corrected)
Results: Mutation induction % of both physical and chemical mutations must be included in results for further studies about the selection of mutagen and its specific concentration.
Response: Done accordingly (mentioned in 2.1. Characterization of M2 populations section)
Where is characterization of M1 generation? It is logical to begin mutation selection from 1st generation instead of 2nd generation. Explain it with scientific reasoning.
Response: Normally the selection of mutations begins from M2 to avoid selecting of morphological mutations occurred due to some physiological changes. The phenotypes appeared due to environmental variation or seedling physiology are disappeared in M2
How homogeneity is maintained in each selected mutant within each selection as it is already mentioned in the introduction that is frequently outcrossed by insects.
Response: Done accordingly (corrected to the performance stability)
Table 2: It is better to write the mutagen name in front of concentration to avoid confusion.
Response: Done accordingly

Round 2
Reviewer 2 Report
Dear author(s),
I have checked your mn and some of my comments have omitted. These are:
Introduction
1. L58, “Chamomile” in the sentence should be the genus (if so), it should be written as the genus name with author name as Matricaria L.
2. L69, scientific name (M. recutita L.) could be written after German chamomile.
3. L110, scientific name of the plant material has already been written in the first mentioned place. After that, it is not necessary in “INTRODUCTION”
Results
4. L121, delete with, at the end of sentence. It should be omitted.
Discussion
5. L274-278, author name should be written when scientific name of the plant species were written where the first mentioned places. Lens culunaris Medik., Vigna mungo (L.) Hepper, Lathyrus sativus L., Hordeum vulgare L.
6. L309, correct Germanchamomile to German chamomile
M&M
7. L316, correct Matricariarecutita L. to Matricaria recutita L.
8. L339-341, M1 and M2 generations especially for M2 cannot be grown in the replicated trials (RCBD), it is meaningless. Please delete or remove this sentence. The selected mutants can be grown in the replicated experiments.
9. L353, correct subtitle as Measured parameters/Studied traits/Studied characteristics
10. L354, remove “above mentioned” and write details of experiments with replication.
Diagram
11. Correct “Irradiated seeds is better” as “Seeds treatments”
12. “Selection” shames or words should be shifted a little bit above. They should be equal with M2 and M3.
13. M2, M3 and M4 should properly be written as M2, M3 and M4.
Conclusions
14. L403, you have already selected good mutants including suitable mechanical harvest, high yield and good oil content. Why did you need additional mutation breeding? An alternative conclusion can be considered as; The selected desirable mutants will directly be used for commercial production after registration.
Author Response
Response to Reviewer 2 Comments (Round 2)
Manuscript ID: plants-1969401
Manuscript Title: Promising mutations for mechanical harvesting, high flowers yield and essential oil content in German chamomile (Matricaria recutita L.) using physical and chemical mutagenesis.
We are very glad that Reviewer 2 highly evaluated our manuscript, and provided constructive comments and valuable suggestions that have helped us further improvement of the quality of our manuscript. All the changes made in response to the Reviewer 2 comments were corrected as track changes in the revised manuscript. We have addressed all of your queries and improved our manuscript following your suggestions as you can see in our point-to-point responses to your comments below.
Comments and Suggestions for Authors
Dear author(s),
I have checked your mn and some of my comments have omitted. These are:
Introduction
- L58, “Chamomile” in the sentence should be the genus (if so), it should be written as the genus name with author name as Matricaria L.
Response: Done accordingly
- L69, scientific name (M. recutita L.) could be written after German chamomile.
Response: Done accordingly
- L110, scientific name of the plant material has already been written in the first mentioned place. After that, it is not necessary in “INTRODUCTION”
Response: Done accordingly
Results
- L121, delete with, at the end of sentence. It should be omitted.
Response: Done accordingly
Discussion
- L274-278, author name should be written when scientific name of the plant species were written where the first mentioned places. Lens culunaris Medik., Vigna mungo (L.) Hepper, Lathyrus sativus L., Hordeum vulgare L.
Response: Done accordingly
- L309, correct Germanchamomile to German chamomile
Response: Done accordingly
M&M
- L316, correct Matricariarecutita L. to Matricaria recutita L.
Response: Done accordingly
- L339-341, M1 and M2 generations especially for M2 cannot be grown in the replicated trials (RCBD), it is meaningless. Please delete or remove this sentence. The selected mutants can be grown in the replicated experiments.
Response: Done accordingly (Corrected)
- L353, correct subtitle as Measured parameters/Studied traits/Studied characteristics
Response: Done accordingly
- L354, remove “above mentioned” and write details of experiments with replication.
Response: Done accordingly
Diagram
- Correct “Irradiated seeds is better” as “Seeds treatments”
Response: Done accordingly
- “Selection” shames or words should be shifted a little bit above. They should be equal with M2 and M3.
Response: Done accordingly
- M2, M3 and M4 should properly be written as M2, M3 and M4.
Response: Done accordingly
Conclusions
- L403, you have already selected good mutants including suitable mechanical harvest, high yield and good oil content. Why did you need additional mutation breeding? An alternative conclusion can be considered as; The selected desirable mutants will directly be used for commercial production after registration.
Response: Done accordingly
